Early cancer screening surveillance in one medical center of China

Yang Ying 1
Du Peng 2
Hou Xiaolu 3
Yan Kun 4
Dai Ying 4
Sun ZhiYing 4
Wu Qi 5
Li Shijie 5
Yan Yan 5
Wang Zhilong 6
Qi Liping 6
Chen Mailin 6
Zheng Hong 7
Gao Weijiao 7
Gao Min 7
Xue Weicheng 8
Zhang Xiaodong 1 zhangxd0829@163.com
1 Key laboratory of Carcinogenesis and Translational Research (Ministry of Education, China), The VIP-II Gastrointestinal Cancer Division of Medical Department, Peking University Cancer Hospital & Institute , Beijing , China
2 Key laboratory of Carcinogenesis and Translational Research (Ministry of Education, China), Department of Urology, Peking University Cancer Hospital & Institute , Beijing , China
3 Western Beijing Cancer Hospital , Beijing , China
4 Key laboratory of Carcinogenesis and Translational Research (Ministry of Education, China), Department of Ultrasonography, Peking University Cancer Hospital & Institute , Beijing , China
5 Key laboratory of Carcinogenesis and Translational Research (Ministry of Education, China), Department of Endoscopy, Peking University Cancer Hospital & Institute , Beijing , China
6 Key laboratory of Carcinogenesis and Translational Research (Ministry of Education, China), Department of Radiology, Peking University Cancer Hospital & Institute , Beijing , China
7 Key laboratory of Carcinogenesis and Translational Research (Ministry of Education, China), Department of Gynecologic Oncology, Peking University Cancer Hospital & Institute , Beijing , China
8 Key laboratory of Carcinogenesis and Translational Research (Ministry of Education, China), Department of Pathology, Peking University Cancer Hospital & Institute , Beijing , China
Hromić-Jahjefendić Altijana
Electronic publication date: 2024 Sep 27
Publication date: 2024
Volume: 12
Electronic Location ID: e18179
Received 2023 Nov 14; Accepted 2024 Sep 4
Copyright: © 2024 Yang et al.
Copyright year: 2024
Copyright holder: Yang et al.
License: This is an open access article distributed under the terms of the Creative Commons Attribution License, which permits unrestricted use, distribution, reproduction and adaptation in any medium and for any purpose provided that it is properly attributed. For attribution, the original author(s), title, publication source (PeerJ) and either DOI or URL of the article must be cited.
License URL: https://creativecommons.org/licenses/by/4.0/

Keywords: Early cancer screening, Family history of malignancy, Sensitivity, Specificity, Positive predictive value

Funding: The authors received no funding for this work.

==============================
Objectives

Cancer screening aims to detect and treat malignant lesions at an early stage and to prolong patients’ lifetime. There is still a lack of effective cancer screening programs in China. We initiated a screening project in 2018 and this study presented the cancer screening status in China.

Methods

We conducted a cross-sectional study in one cancer-care medical center of China. The screening program included routine blood tests, plasma tumor markers, gastric endoscopy, colonoscopy, ultrasound, and computed tomography (CT) scans. Screening results were presented as sensitivity, specificity and positive predictive values (PPVs).

Results

Twenty-three (1.46%) out of 1,576 participants were eventually diagnosed with malignant tumors or high-grade intraepithelial neoplasia (HGIN). A family history of malignancy (78.26% in diagnosed cancer and HGIN vs. 46.36% in the others) was the only statistically significant parameter associated with cancer detection (p = 0.002). None of the common tumor markers were associated with the cancers screened. Except for colonoscopy (50.00%) and ultrasound for renal cancer (66.67%), the sensitivities of most screening methods were 100%. The specificities of all the screening means were above 96%. Most PPVs ranged from 30–60%.

Conclusion

We emphasized risk stratification for early cancer screening, such as a family history of cancer. The survey illustrated that gastric endoscopy, colonoscopy, ultrasound, and lung CT for early cancer screening had high specificity, reasonable sensitivity, and PPV. We anticipated this report would motivate larger-sample studies to estimate the risk-to-benefit ratio of cancer screening and urge the establishment of a native Chinese screening project and even guidelines.

Introduction

The most updated data (Chen et al., 2016) show that there were an estimated 4,292,000 cancer incident cases in China in 2015. In males, five cancers, including cancers of the lung and bronchus, stomach, esophagus, liver, and colorectum, accounted for nearly two-thirds of all cancer cases. Among women, the five corresponding cancers were breast, lung and bronchus, stomach, colorectum, and esophagus, accounting for nearly 60% of all cases. It was estimated that approximately 2,814,000 Chinese people died of cancer in 2015. The five leading causes of cancer death among both men and women were cancers of the lung and bronchus, stomach, liver, esophagus, and colorectum, accounting for approximately three-quarters of all cancer-related deaths. Only 36.9% of cancer patients in China survived for more than 5 years after diagnosis.

Early cancer screening aims to prolong the lifetime by detecting malignant lesions at their precursor or early stage, which could prevent more invasive lesions and might be cured by effective treatment. Taking colorectal cancer as an example, benign polyps may progress to malignant colorectal lesions and can be detected and eradicated by endoscopy. A pooled analysis of three randomized trials on sigmoidoscopy screening revealed that the risk of colorectal cancer was up to 25% lower in the screening group with 10–12 years of follow-up (Holme et al., 2017). Updated estimates showed a 31% decrease in colorectal cancer incidence and a 50% decrease in colorectal cancer–related deaths due to colonoscopy (Bretthauer et al., 2022). In terms of lung cancer, the UK Lung Cancer Screening Trial showed that 75% of detected lung cancers were at stage I with a satisfactory prognosis (Field et al., 2016). No association was observed between endoscopic screening and the risk of gastric cancer (Zhang et al., 2018); however, it was associated with a 40% reduction in cancer-related deaths (Kim & Cho, 2021). Therefore, colonoscopy and chest computed tomography (CT) have been established as valuable tools for the early detection and lifesaving of colorectal cancer and lung cancer (Bretthauer et al., 2022; Field et al., 2016). Nationwide gastroscopy screening has only been conducted in Japan and Korea (Leung et al., 2008).

In China, systematic early cancer screening programs and official guidelines for professionals are still lacking. Thus, our center initiated a cancer screening project in 2018, and more than 1,000 individuals received screenings.

Methods

Since our center established the early cancer screening program in July 2018, with media promotion, the general public had volunteered to participate. Those with urgent medical conditions or synchronous cancers were excluded from the program. However, participants with either carcinoma in situ or early-stage cancer, who had been diagnosed as ‘cured’ by their oncologists, could be included. People who refused to share their health data were excluded from the statistical analyses. All others were included in the program and the subsequent analyses. Until June 2022, a total of 1,576 individuals had been recruited for the study.

The screening program included two-day medical examination and testing. On the first day, general characteristics were collected via questionnaires including age, sex, family cancer history, occupation and marital status; smoking, drinking, chronic disease history, etc. We recorded participants’ height and weight, blood pressure and heart rates. Participants were asked to present fasting, and routine blood tests, including complete blood count and blood biochemistry tests, were performed. Plasma tumor markers were also included in the test at most participants’ requests and for research purposes. After blood drawn, participants received CT scans and ultrasonography. The next day participants received gastric endoscopy and colonoscopy. Additional laboratory tests and imaging, such as magnetic resonance imaging (MRI), were performed at participants’ discretion. The participants’ recruitment was seen in Fig. 1.

Figure 1 Participants’ recruitment flowchart.

For suspicious renal lesions, enhanced CT or MRI were applied as confirmed diagnosis. For mammary ultrasound Breast Imaging Reporting and Data System (BI-RADS) 4 lesions, biopsy was recommended, however, MRI was an acceptable alternative to biopsy. In terms of lung lesions found by CT, biopsy or surgery was recommended, but intensive follow-up of a ground-glass nodule that was less than 1 centimeter was allowable. For all other lesions detected through screening tests, histopathological diagnosis served as the gold standard. The following lesions strongly required pathological diagnosis: (1) suspicious malignant lesions detected by gastroscopy; (2) suspicious malignant lesions, adenomas or polyps detected by colonoscopy; (3) ultrasound BI-RADS 5 lesions; (4) ultrasound Thyroid Imaging Reporting and Data System (TI-RADS) 4 or above lesions; (5) suspicious malignant lesions of gastrointestinal or ovarian detected by CT.

The confirmed diagnoses, the majority of which were pathological diagnoses, were defined as true positives. No lesions were found by the screening method or suspected lesions ruled out by the gold standard diagnostic methods were treated as true negatives. Lesions found by screening tests but ruled out by standard diagnostic methods were considered false positives. Lesions not detected by screening tests but diagnosed by confirmed diagnostic instruments were considered false negatives. The screening means were valued by sensitivity, specificity, positive predictive value (PPV) and negative predictive value (NPV). Statistical significance was set at a two-sided p value ≤ 0.05. The independent two-sample t-test was conducted to compare the two group means of continuous variables; however, if the data did not conform to normal distribution or the variances between the two groups were not equal, the Mann-Whitney U test was used. The comparison of categorical variables between two groups adopted the χ2 test or Fisher’s exact test, whenever appropriate.

All the participants signed a written agreement to publish their screening results without recognizable identities for scientific purposes. Ethics approval was waived by the Institutional Review Board (IRB) of Peking University Cancer Hospital and Institute (IRB number 2017YJZ57), as all screening tests were routine clinical examinations without extra benefits or harm.

The datasets generated and analyzed during the current study are not publicly available for patient privacy protection. All participants were not involved in the design, conduct, reporting, or dissemination plans of our research.

This was a cross-sectional study without blinding, randomization, or power analysis. The screening results were presented as counts and percentages.

Results

A total of 1,576 individuals decided to participate in early cancer screening between July 2018 and June 2022. There were 702 (44.5%) male participants and 874 (55.5%) female participants in the study, with a median age of 47 (range: 17–81); and the mean age with a standard deviation was 46.9 ± 12.2. Twenty-two (1.40%) patients were eventually diagnosed with malignant tumors. Specifically, there were seven (30.43%) thyroid papillary carcinomas, five (21.74%) lung adenocarcinomas, two (8.70%) breast cancers, two (8.70%) renal clear cell carcinomas, one colon adenocarcinoma, one esophageal squamous cell carcinoma, one gastric adenocarcinoma, one ovarian adenocarcinoma, and one leukemia; there was also one with synchronous primary cancers of both renal clear cell carcinoma and lung adenocarcinoma. Additionally, one patient had high-grade colon intraepithelial neoplasia (HGIN). The results were presented in Table 1.

Table 1 Diagnosed cancers and high-grade intraepithelial neoplasia (HGIN) through the early cancer screening.

Confirmed cancers and HGIN (N = 23)	N	%	
Thyroid papillary carcinoma	7	30.43	
Lung adenocarcinoma	5	21.74	
Breast cancer	2	8.70	
Renal clear cell carcinoma	2	8.70	
Colon adenocarcinoma	1	4.35	
Esophageal squamous cell carcinoma	1	4.35	
Gastric adenocarcinoma	1	4.35	
Ovarian adenocarcinoma	1	4.35	
Leukemia	1	4.35	
Renal clear cell carcinoma and lung adenocarcinoma	1	4.35	
Colon HGIN	1	4.35	

We compared the general characteristics between patients with confirmed cancer, including one colon HGIN (as cases) and the others screened (as controls). Individuals diagnosed with cancer and HGIN were slightly older, with a mean age of 48.8 years, and the mean age of the controls was 46.9 years. There were slightly more women (65.22%) among the detected cases than among the controls (55.31%). More patients had a previous cancer history (8.70%), while this proportion was only 3.80% in the controls. The distribution patterns of smoking, drinking, and secondhand smoke were similar in both the cases and their counterparts. A family history of malignancy was the only statistically significant parameter associated with a cancer diagnosis by screening (p = 0.002). In the control group, nearly half (46.36%) had a family history of malignant tumors, while 78.26% had a family history of cancer in the cases. The results were presented in Table 2.

Table 2 Association between general characteristics and detected high-grade intraepithelial neoplasia (HGIN) and cancer.

Clinical characteristics	Confirmed cancers and HGIN	Without confirmed cancers	P-value	
	N (%)	N (%)		
Age				
Median (Mean ± Std)	48 (48.8 ± 9.1)	47 (46.9 ± 12.3)	0.450	
<20	0 (0)	3 (0.19)		
20–29	0 (0)	108 (6.95)		
30–39	4 (17.39)	403 (25.95)		
40–49	8 (34.78)	368 (23.70)		
50–59	8 (34.78)	413 (26.59)		
60–69	3 (13.04)	212 (13.65)		
>70	0 (0)	46 (2.96)		
Gender			0.343	
Male	8 (34.78)	694 (44.69)		
Female	15 (65.22)	859 (55.31)		
Family history of cancer			0.002	
Yes	18 (78.26)	720 (46.36)		
No	5 (21.74)	833 (53.64)		
History of cancer			0.223	
Yes	2 (8.70)	59 (3.80)		
No	21 (91.30)	1,494 (96.20)		
Smoking			1.000	
Yes	4 (17.39)	296 (19.06)		
No	19 (82.61)	1,257 (80.94)		
Drinking			0.518	
Yes	5 (21.74)	432 (27.82)		
No	18 (78.26)	1,121 (72.18)		
Secondhand smoke			0.865	
Yes	5 (21.74)	361 (23.25)		
No	18 (78.26)	1,192 (76.75)		

There were no statistically significant differences in tumor markers between cases and non-cases. Most tumor markers, such as carcinoembryonic antigen (CEA), carbohydrate antigen 19-9 (CA19-9), carbohydrate antigen 125 (CA125), and cytokeratin fragment 21-1 (CYFRA21-1), were elevated in less than 2% of non-cancer participants. Meanwhile, high carbohydrate antigen 72.4 (CA72.4) was detected in 8.50% of the controls, with no statistical significance. The results were presented in Table 3.

Table 3 Association between tumor markers and detected high-grade intraepithelial neoplasia (HGIN) and cancers.

Tested tumor markers	Confirmed cancers and HGIN
N (%)	Without confirmed cancers
N (%)	P-value	
CEA			0.057	
Abnormal	2 (8.70)	25 (1.61)		
Normal	21 (91.30)	1,528 (98.39)		
CA125			0.278	
Abnormal	1 (4.35)	21 (1.35)		
Normal	22 (95.65)	1,532 (98.65)		
CA19-9			0.310	
Abnormal	1 (4.35)	24 (1.55)		
Normal	22 (95.65)	1,529 (98.45)		
CA72.4			1.000	
Abnormal	2 (8.70)	132 (8.50)		
Normal	21 (91.30)	1,421 (91.50)		
AFP			1.000	
Abnormal	0 (0)	1 (0.06)		
Normal	23 (100)	1,552 (99.94)		
NSE			1.000	
Abnormal	0 (0)	11 (0.71)		
Normal	23 (100)	1,542 (99.29)		
CYFRA21-1			1.000	
Abnormal	0 (0)	26 (1.67)		
Normal	23 (100)	1,527 (98.33)		
CA242			1.000	
Abnormal	0 (0)	1 (0.06)		
Normal	23 (100)	1,552 (99.94)		
Note:

CEA, carcinoembryonic antigen; CA125, carbohydrate antigen 125; CA19-9, carbohydrate antigen 19-9; CA72.4, carbohydrate antigen 72.4; AFP, alpha-fetoprotein; NSE, neuron-specific enolase; CYFRA21-1, cytokeratin fragment 21-1; CA242, carbohydrate antigen 242.

Patients might refuse any examination, and the dropout rates of gastric endoscopy, colonoscopy and lung CT were 28.43%, 31.60% and 27.41%, respectively. The ultrasound attrition rates of abdomen, thyroid, breast, and female pelvis, prostate was 11.36%, 9.77%, 10.64%, 12.36% and 20.23%, correspondingly. Of the 1,128 individuals who underwent gastric endoscopy, six (0.53%) were suspected of having malignant lesions, of whom two (33.33%) were confirmed to have malignancies. Three of 1,078 (0.28%) individuals undergoing colonoscopy were suspected to have malignant lesions, and one (33.33%) was pathologically confirmed as adenocarcinoma. One of the 369 (0.37%) nonadenomatous polyps was confirmed to be HGIN. This could be seen in Table 4.

Table 4 Screening program results.

	N	%	Dropout (%)	
Gastric endoscopy				
Suspected malignant	6	0.53		
Pathology confirmed adenocarcinoma	1 (16.67)			
Pathology confirmed squamous carcinoma	1 (16.67)			
Pathology confirmed benign lesions	4 (66.67)			
Chronic atrophic gastritis/Barrett’s Esophagus	269	23.85		
Other benign lesions	694	61.52		
No abnormal findings	159	14.10		
Missing	448		28.43	
Colonoscopy				
Suspected malignant	3	0.28		
Pathology confirmed adenocarcinoma	1 (33.33)			
Pathology confirmed adenoma	2 (66.67)			
Pathology confirmed other benign lesions	0 (0)			
Adenoma	48	4.45		
Pathology confirmed adenocarcinoma	0 (0)			
Pathology confirmed adenoma	30 (85.71)			
Pathology confirmed other benign lesions	5 (14.29)			
Without pathology diagnosis	13			
Non-adenomatous polyps	369	34.23		
Pathology confirmed HGIN*	1 (0.37)			
Pathology confirmed adenoma	162 (59.56)			
Pathology confirmed other benign lesions	109 (40.07)			
Without pathology diagnosis	97			
Others	658	61.04		
Missing	498		31.60	
Prostate ultrasound				
Abnormal (hyperplasia, calcification, nodule)	371	66.25		
Normal	189	33.75		
Missing	142		20.23	
Female pelvis ultrasound				
Suspected malignant ovarian lesion	6	0.78		
Pathology confirmed ovarian cancer	1 (16.67)			
Metastatic breast cancer	1 (16.67)			
Pathology confirmed benign lesions	4 (66.67)			
Hysteromyoma	194	25.33		
Ovarian cyst	59	7.7		
Hysteromyoma with ovarian cyst	25	3.26		
Teratoma	2	0.26		
No abnormalities	480	62.66		
Missing	108		12.36	
Abdomen ultrasound				
Renal lesion	9	0.64		
Pathology confirmed renal cancer	2 (22.22)			
Malignant lesions ruled out	7 (77.78)			
Liver lesion	3	0.21		
Pancreas lesion	2	0.14		
Suprarenal gland lesion	1	0.07		
Peritoneal and retroperitoneal lesion	2	0.14		
Liver cirrhosis	6	0.43		
Gallbladder polyps	66	4.72		
No abnormalities	1,308	93.63		
Missing	179		11.36	
Mammary ultrasound				
BI-RADS**	2	0.26		
Pathology confirmed malignant	1 (50.00)			
Pathology confirmed benign	1 (50.00)			
BI-RADS 4	30	3.84		
Pathology confirmed malignant	1 (3.33)			
Malignant lesions ruled out	29 (96.67)			
Other	749	95.90		
Missing	93		10.64	
Thyroid ultrasound				
TI-RADS*** 4 and higher	36	2.53		
Pathology confirmed malignant	7 (19.44)			
Pathology confirmed benign	29 (80.56)			
Other	1,386	97.47		
Missing	154		9.77	
CT**** of lung				
Suspected lung cancer	15	1.31		
Pathology confirmed malignant	6 (40.00)			
Malignant lesions ruled out	9 (60.00)			
Suspected metastasis cancer	4	0.35		
Ground-glass opacity	98	8.57		
Other	1,027	89.77		
Missing	432		27.41	
CT of abdomen and pelvis				
Suspected liver lesions	2	0.21		
Suspected colorectal lesions	1	0.10		
Pathology confirmed malignant	0 (0.00)			
Pathology confirmed benign	1 (100.00)			
Suspected renal lesion	5	0.52		
Pathology confirmed malignant	3 (60.00)			
Malignant lesions ruled out	2 (40.00)			
Suspected ovarian lesion	1	0.1		
Pathology confirmed malignant	1 (100.00)			
Suspected pancreas lesion	1	0.1		
Suspected bone lesion	1	0.1		
Other	943	98.85		
Missing	622		39.47	
Notes:

* HGIN, high-grade intraepithelial neoplasia.

** BI-RADS, Breast Imaging Reporting and Data System.

*** TI-RADS, Thyroid Imaging Reporting and Data System.

**** CT, computed tomography.

In terms of ultrasound examinations, seven thyroid cancers were detected out of thirty-six TI-RADS four and higher cases (19.44%), while one breast cancer was detected in two BI-RADS five cases (50.00%), and another breast cancer was diagnosed among thirty BI-RADS four cases (3.33%). Additionally, 766 female participants underwent pelvic ultrasound, and 6 (0.78%) had suspected malignant lesions, of whom 2 (33.33%) were confirmed to be malignant by pathological diagnosis. Abdominal ultrasound revealed nine renal lesions, two of which were confirmed as cancers (22.22%), while abdominal CT revealed five renal lesions and three were confirmed as malignancies (60.00%). Notably, only one case was suspected to be malignant colon cancer on abdominal CT and was ruled out by further examination. Lung CT scanning was conducted in 1,144 participants, and six lung cancers were diagnosed among 15 suspected cases (40.00%). Table 4 presented the results.

Table 5 presents the sensitivity, specificity, PPV, and NPV of each screening item. The specificities of all screening methods were above 96%; NPVs were above 99%, and except for colonoscopy (50.00%) and ultrasound for renal cancer (66.67%), the sensitivities of most screening means were 100%. The majority of the PPVs ranged from 30–60%. The PPVs of ultrasound for breast, thyroid, and renal cancers were relatively low at 6.25%, 19.44%, and 22.22%, respectively.

Table 5 Sensitivity, specificity, positive predictive value and negative predictive value of different screening program and type-specific cancer.

	True positive	True negative	Sensitivity (%)	Specificity (%)	Positive predictive value (%)	Negative predictive value (%)	
Endoscopy for upper gastrointestinal cancer	
Test positive	2	4					
Test negative	0	1,122	100.00	99.64	33.33	100.00	
Endoscopy for colorectal cancer	
Test positive	1	2					
Test negative	1	1,074	50.00	99.81	33.33	99.91	
Ultrasound for ovarian cancer	
Test positive	2	4					
Test negative	0	760	100.00	99.48	33.33	100.00	
Ultrasound for renal cancer	
Test positive	2	7					
Test negative	1	1,387	66.67	99.50	22.22	99.93	
Ultrasound for breast cancer	
Test positive	2	30					
Test negative	0	749	100	96.15	6.25	100	
Ultrasound for thyroid cancer	
Test positive	7	29					
Test negative	0	1,386	100.00	97.95	19.44	100.00	
CT* for lung cancer	
Test positive	6	9					
Test negative	0	1,129	100.00	99.21	40.00	100.00	
CT for renal cancer	
Test positive	3	2					
Test negative	0	949	100.00	99.79	60.00	100.00	
Note:

* CT: computed tomography.

Discussion

There are differences in medical resources, accessibility to services, health policies, and other factors in various countries and regions, the guidelines for early cancer screening around the world vary. The American Cancer Society recommends that women aged 40 to 44 may choose to start having annual mammograms, and women aged 45 to 54 should continue to have mammograms annually. Women aged 55 and older can have a mammogram every 2 years (Oeffinger et al., 2015). A systematic review on breast cancer screening guidelines published by different countries or regions between 2010 and 2021 found that the age group of 50 to 69 was the best for screening (Ren et al., 2022). Both the United States and Australia suggest regular screening for colorectal cancer starting at the age of 45, and up to the age of 75, and colonoscopy is recommended (Kirby, 2023; Lin et al., 2021). In China, it is recommended to conduct lung cancer screening using low-dose spiral CT (LDCT) among people aged 50 to 74 (Chinese Expert Group on Early Diagnosis and Treatment of Lung Cancer, 2023). The American Cancer Society recommends annual lung cancer screening for people aged 50 to 80 with LDCT, particularly for the high-risk population of current or former smokers (Wolf et al., 2024). Except for the expert consensus on lung cancer screening (Chinese Expert Group on Early Diagnosis and Treatment of Lung Cancer, 2023), we are still lacking screening guidelines for medical professionals. Our medical center established this screening program to explore early cancer screening status in China.

In our study, 23 out of 1,576 participants were diagnosed with cancer, including one case of HGIN, which accounted for 1.46% of all participants without specific cancer type designations. Our study showed that nearly 70% of the screened cancer cases were individuals aged 40–60 years of age, which was consistent with the age ranges of most screening guidelines. Among these 23 patients with cancer and HGIN, the top four detected cancer types were thyroid papillary carcinoma (7/1,422, 0.49%), lung adenocarcinoma (6/1,144, 0.52%), renal clear cell carcinoma (3/1,397, 0.21%), and breast cancer (2/781, 0.26%). Due to the availability and convenience of ultrasound tests in our center, we detected a significant number of thyroid papillary carcinomas and renal cancers; also, one case of renal cancer that was missed on ultrasound but captured by CT scans. However, the prognosis of these two types of cancer is generally favorable, and whether early screening for thyroid papillary carcinoma and renal cancer would reduce mortality without extra harm is controversial (Ahn, Kim & Welch, 2014; Rosiello et al., 2021; Usher-Smith et al., 2020). Therefore, the results required careful scrutiny, and we did not recommend routine screening for thyroid papillary carcinoma and renal cancer yet. Regarding lung and breast cancer, which is recommended by most screening guidelines, a previous study indicated that the detection rate of low-dose CT for lung cancer in China was approximately 1.68% (Zhao & Wu, 2015). In Canada, approximately 0.37% of invasive breast cancer cases were detected in women aged 50 to 69 years (Seely & Alhassan, 2018). The detection rate in our study was relatively low, and the possible reasons for this were multifaceted. For instance, it might be due to a lack of high-risk population selection, or it could be attributed to a refusal rate of lung CT as high as 27.41%. Additionally, patients with lung or breast cancer might have undergone screening at primary care units rather than at our center.

Screening for high-risk individuals is the best way to maximize the cost-effectiveness of screening. Family history is a well-established risk factor for cancer screening, especially for breast and colorectal cancers (Montminy et al., 2020; Ren et al., 2022). In our study, only a family history of cancer was significantly associated with increased cancer detection. Our findings further emphasized the importance of screening individuals with a family history of cancer. However, our sample size was insufficient for stratification by cancer type.

Our study showed higher sensitivity, specificity and PPVs than those of previous studies. Previous reports showed that the sensitivity of cancer screens mainly ranged from 70% to 80%, and the specificity was approximately 60% to 70% (Schiffman, Fisher & Gibbs, 2015); the PPV was approximately 20% for mammograms (Moshina et al., 2016) and ranged from 10% to 20% for lung cancer (Krilaviciute & Brenner, 2021). Our results showed that the sensitivity and specificity for the majority of the cancers we studied were above 96%, and the PPVs were approximately 30–60%. These might be attributed to our state-of-the-art equipment and the expertise of our highly proficient physicians. The only exception was that the PPV of breast ultrasound was only 6.25%, which was lower than the PPV of mammography in Western countries. This might be related to ethnicity differences in breast structure, as well as the differences in imaging mechanisms between ultrasound and mammography. Considering the widespread use of ultrasound in China and additional radiation risks, we still regarded breast ultrasound as an effective method for screening.

It was widely accepted that blood tumor markers were good indicators of cancer treatment response and tumor burdens; however, their implications for early cancer detection were not definitive, and it was not recommended to include tumor markers in routine early cancer screening programs (Loud & Murphy, 2017; Schiffman, Fisher & Gibbs, 2015). A large number of participants asked for this laboratory examination, we therefore conducted tests for surveillance. Elevated CA72.4 levels were found in approximately 8.50% of participants, and other common tumor markers that were higher than the upper normal limits were found in less than 2% of participants. Neither CA72.4 nor the other markers were associated with the cancer detection rate. We suggested close follow-up to participants with elevated tumor markers. However, most participants had tumor marker re-examinations in their nearby medical facilities and we lost the follow-up data. It was still questionable whether tumor markers had indicative significance for early-stage cancer, which markers should be recommended, and what the follow-up frequency should be; these issues required further study and exploration.

Overdiagnosis and overtreatment are major concerns during screening. Participants may not receive a survival benefit from screening but may have extra harm, especially from invasive examinations, radiation exposure, and unnecessary therapy, such as prostate cancer and ovarian cancer (Fleshner, Carlsson & Roobol, 2017; Menon et al., 2021). In China, female participants are more likely to undergo ultrasound screening for breast cancer. Similar to mammography, ultrasound has high false-positive rates. In our study, the false-positive rate was as high as 93.75%, which would lead to additional MRIs or needless biopsies.

The study suffered from several limitations. Firstly, as an observational study, we did not establish the inclusion and exclusion criteria as strict as clinical trials. Participants volunteered for early cancer screening and might request or decline any examinations and tests, which resulted in an approximate 30% dropout rate for gastric endoscopy, colonoscopy, or lung CT; and around 10% dropout rate for ultrasound. The unavoidable selection bias led to the results being either underestimated or overestimated. Secondly, this was a cross-sectional study with its inherent characteristics. For example, the general information collection, including family cancer history, smoking and drinking, was self-reported via questionnaire and suffered from recall bias. Also, the screening relied heavily on the techniques, which made measurement bias inevitable. These biases would lead the results toward either direction. Moreover, a cross-sectional study was without temporal sequence, which meant the study was rarely affected by confounding factors but did not allow for the evaluation of prognosis improvements, therefore, we only presented the efficiency, rather than the effectiveness. Finally, considering it was a study on cancer screening, the sample size was relatively small and did not qualify for stratification by cancer type, moreover, our medical center boasted top-tier screening technology, complemented by the expertise of our adept medical staff, all of which made the generalization of our results questionable.

Conclusion

Both the cancer incidence and mortality rates are high in China. Many cancers can be cured at early stages. We initiated a project to screen for the most common cancers in the Chinese population. Family history was the only significant risk factor for increased cancer detection. Based on our results, we recommended gastric endoscopy, colonoscopy, breast ultrasound, and lung CT for regular cancer screening. It is worth noting that approximately 30% of participants refused to undergo gastric endoscopy, colonoscopy, or lung CT. Due to their reasonable PPVs as well as the widely recognized benefits of screening, it is crucial to alleviate public concerns and encourage the uptake of gastrointestinal endoscopy and lung CT scans in the general population. However, as a specialized medical institution, our screening technology was at the forefront, and our medical staff were highly competent. The study was also limited by sample size and study design. Thus, caution should be exercised when generalizing our results. We hope that this report will motivate larger-sample studies to estimate the risk-to-benefit ratio of cancer screening and promote the establishment of our native screening project and the development of guidelines.

Ethics approval and consent to participate

All participants signed a written informed consent form before undergoing the screening programs and agreed to publish their screening results without recognizable identities for scientific intention. Ethics approval was waived by the Institutional Review Board of Peking University Cancer Hospital and Institute since all screening tests were routine clinical examinations without extra benefits or harm.

Supplemental Information

Supplemental Information 1 Raw data and codes for all analyses.

Supplemental Information 2 Codebook for raw dataset.

We thank all our participants for their contribution to this work.

Additional Information and Declarations

Competing Interests

Author Contributions

Human Ethics

Data Availability

The authors declare that they have no competing interests.

Ying Yang conceived and designed the experiments, analyzed the data, prepared figures and/or tables, authored or reviewed drafts of the article, and approved the final draft.

Peng Du performed the experiments, analyzed the data, authored or reviewed drafts of the article, and approved the final draft.

Xiaolu Hou analyzed the data, prepared figures and/or tables, authored or reviewed drafts of the article, and approved the final draft.

Kun Yan performed the experiments, authored or reviewed drafts of the article, and approved the final draft.

Ying Dai performed the experiments, authored or reviewed drafts of the article, and approved the final draft.

ZhiYing Sun performed the experiments, authored or reviewed drafts of the article, and approved the final draft.

Qi Wu performed the experiments, authored or reviewed drafts of the article, and approved the final draft.

Shijie Li performed the experiments, authored or reviewed drafts of the article, and approved the final draft.

Yan Yan performed the experiments, authored or reviewed drafts of the article, and approved the final draft.

Zhilong Wang performed the experiments, authored or reviewed drafts of the article, and approved the final draft.

Liping Qi performed the experiments, authored or reviewed drafts of the article, and approved the final draft.

Mailin Chen performed the experiments, authored or reviewed drafts of the article, and approved the final draft.

Hong Zheng performed the experiments, authored or reviewed drafts of the article, and approved the final draft.

Weijiao Gao performed the experiments, authored or reviewed drafts of the article, and approved the final draft.

Min Gao performed the experiments, authored or reviewed drafts of the article, and approved the final draft.

Weicheng Xue performed the experiments, authored or reviewed drafts of the article, and approved the final draft.

Xiaodong Zhang conceived and designed the experiments, analyzed the data, authored or reviewed drafts of the article, and approved the final draft.

The following information was supplied relating to ethical approvals (i.e., approving body and any reference numbers):

The Institutional Review Board of Peking University Cancer Hospital and Institute approved this research (IRB number 2017YJZ57).

The following information was supplied regarding data availability:

The raw measurements are available in the Supplemental Files.

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
