# Peer review of "Early cancer screening surveillance in one medical center of China"

_PeerJ, doi:10.7717/peerj.18179_

## Round 0.1 · original submission · Major Revisions

Provide detailed information on the inclusion and exclusion criteria, recruitment process, and characteristics of the study participants.
Clarify the timing and distribution of various tests conducted.
Specify whether the p-value is two-sided or one-sided.
Clarify which t-test was used and under what conditions the Mann-Whitney U test was applied.
Define true positive, false positive, false negative, and true negative for each type of cancer/test.
Discuss the limitations of the study, including concerns about generalizability and potential biases.
Analyze changes in blood tumor markers in diagnosed cancer patients and compare with other studies.
Add a horizontal line in Table 1 to separate "HGIN" from "Confirmed Cancers."
Add a conclusion to summarize key findings, implications, and future research directions.

Reviewer 1 ·

Basic reporting

no comment

Experimental design

-The inclusion and exclusion criteria of the study participants are not quite clear. Please provide more information on this early cancer screening program that the 1576 people participated in. How these people were recruited? What are the inclusion and exclusion criteria of this program? When these participants received different tests such as gastric endoscopy, colonoscopy, ultrasonography, and CT scans? How many participants received each test? A flow chart would help clarify the process.
- Please also provide some information on this one medical center where the study was conducted. What are the characteristics of the patient population in this medical center? How representative is this center of all the medical centers in China or Beijing?
- In line 83, please clarify whether the p-value is two-sided or one-sided.
- In line 84, please clarify which t-test. The Student’s independent two-sample t-test? Please explain in which situation when the Mann-Whitney U test or t-test was used respectively.
- In lines 82-83, please clarify how the true positive, false positive, false negative, and true negative were defined for each type of cancer/test in your study.

Validity of the findings

- Since this study was conducted in one medical center and only ~ 20 people were diagnosed with a few types of cancers. The generalizability of your study results is quite concerning. Could you provide more discussion on this limitation?
- Please add more study limitations to the discussion section, such as generalizability, patient self selection bias, etc.

·

Basic reporting

no comment

Experimental design

no comment

Validity of the findings

no comment

Additional comments

It has been a pleasure reviewing this paper. I appreciate the time and effort the researchers put into this survey study. The writing and objective of this article are clear and concise. The authors have done a great job on this. The use of appropriate statistical tests, including the t-test, chi-squared test, Fisher's exact test, Mann-Whitney U test, along with well-reported statistical measures, such as sensitivity, specificity, positive predictive value, and negative predictive value, enhances the reliability of the results presented. The manuscript responsibly addresses its limitations, appropriately cautioning against the generalization of the results, which is a sign of rigorous scientific reporting. The informative discussion offers valuable insights and recommendations for specific screening methods, which could benefit the field.

My only comment is that the manuscript currently lacks a concluding section, which is essential for summarizing the study's key findings, implications, and future directions. A well-articulated conclusion would not only provide a succinct summary of the work done but also reinforce the importance of the findings and help clearly state the next steps or potential areas for further research.

Overall, the manuscript is of high quality and merits publication after addressing the noted drawback. Including a conclusive section would complete the narrative and enhance the manuscript's utility to its readers.

Reviewer 3 ·

Basic reporting

The authors performed cancer screening in 1576 patients and presented the result here, thus potentially benefiting future studies and patients. Yet, I would really appreciate some additional discussions, as detailed below.

Experimental design

(1) Prior to presenting the result, it is recommended to provide some description of how the 1576 patients were selected. In other words, what motivated the patients to participate in this study? Is there any possible bias there? Hopefully, with this information, our future readers can better understand “a lack of high-risk population selection,” as discussed in Line 132 and similar. Also, a table quickly summarizing the patients’ demographic characteristics, such as age, sex, past medical history, etc., before dividing them into two groups would be really appreciated.

Validity of the findings

(2) In the discussion section, is it possible to present the percentages of various cancers screened and diagnosed here and compare those percentages with similar information from other studies (e.g., expanding the first paragraph in the Discussion section)?
(3) In those patients diagnosed with cancer, do they show any changes in any of the tested blood tumor markers? Is this result consistent with other studies focusing on the particular cancer type and the relevant blood tumor marker(s)? For those having abnormal blood tumor markers but without cancer, were the levels significantly higher than the normal range? Was there any follow-up visit performed or planned? It seems a little premature to advise against the blood tumor marker tests.
(4) Concerning Table 1, kindly add a horizontal line separating the bottom row (HGIN) from the rest (Confirmed Cancers).
(5) For the 18 patients having a family history of cancer and diagnosed with cancer here, do we happen to know if the cancer types are the same?
(6) For the 61 (2+59) patients having a history of cancer, do we happen to know how long they have been free of cancer at the time of screening and what type of cancers they had?
(7) Is it possible to briefly summarize cancer screening(s) advised in other countries if there is no such information in China?

Additional comments

(8) It is suggested that “their lifetime” in Line 25 be changed to “patients’ lifetime” for clarity.
(9) For consistency, it is suggested to change the current tense in Lines 73 and 82 to the past tense, if appropriate.

---

## Round 0.2 · Minor Revisions

Please can you respond to the important comments of Reviewer 3.

Reviewer 1 ·

Basic reporting

The authors have sufficiently addressed my concerns. I have no further comments.

Experimental design

The authors have sufficiently addressed my concerns. I have no further comments.

Validity of the findings

The authors have sufficiently addressed my concerns. I have no further comments.

Reviewer 3 ·

Basic reporting

Many thanks for the authors’ amendments. Much appreciated. I only have one remaining concern, which seems critical for arriving at the claimed conclusion. Kindly see more details below.

Experimental design

Concerning lines 102 and 103, I assume all of the lesions were found by a screening test. If this understanding is correct, what’s the difference(s) between true negatives and false positives? Should true negatives be defined as no lesion and no diagnosis by a standard method?

Further along this line, it seems the current study design does not allow an accurate evaluation of true negatives and false negatives since if there is no lesion detected, the standard diagnosis method is not recommended. In other words, we do not really know if the negatives are truly negative. Accordingly, are the presented sensitivity (True Positives / (True Positives + False Negatives)), specificity (True Negatives / (True Negatives + False Positives)), and NPV (True Negatives / (True Negatives + False Negatives)) calculations accurate, reasonable, and not misleading? It is noted that PPV, which is True Positives / (True Positives + False Positives), seems to remain valid.

Validity of the findings

No comments.

---

## Round 0.3 · accepted · Accept

I confirm that the Authors have addressed all of the reviewers' comments